

# Effectiveness of modifications to preadjusted appliance prescriptions based on racial dental characteristics assessed by the ABO Cast-Radiograph Evaluation: A propensity score matching study

Yanhao Chu, Lingling Zhang, Yatao Zhao, Fang Yi and Yanqin Lu

Department of Orthodontics, Xiangya School of Stomatology, Hunan Key Laboratory of Oral Health Research, Xiangya Stomatological Hospital, Central South University, Changsha, Hunan, China

## ABSTRACT

**Background**. Because racial discrepancies in dental characteristics are known to exist, designing preadjusted appliances according to racial normal occlusion data would be expected to improve treatment results. However, whether modifications based on racial characteristics can improve treatment outcomes in the clinic remains to be investigated.

**Methods**. To study the influence of prescription type on treatment outcomes, 91 patients treated with Chinese or Roth prescription appliances were selected as an initial sample. Two groups of patients were selected by propensity score matching (1:1) to limit the effects of confounding factors, including age, sex, case complexity, and extraction plan. Discrepancy Index and cervical vertebral maturation values were used to quantify case complexity and patient age, respectively. After matching, the final sample of 60 patients consisted of two groups of 30 patients each: group 1 had been treated with a Chinese prescription appliance and group 2 had been treated with a Roth prescription appliance. ABO casts and radiograph evaluation (CR-Eval) and lateral cephalograms were utilized to compare the treatment outcomes of the two groups.

**Results**. The total ABO scores of groups 1 and 2 were 22.03 and 23.87, respectively. There were no significant differences between the two groups in total ABO score or in seven other sub-scores; however, there was a significant difference between the two groups in mandibular canine alignment score.

**Conclusions**. There are no significant differences in overall treatment outcomes between the Chinese and Roth prescription appliances. The Chinese prescription yielded better alignment results in the mandibular canine for Chinese patients.

# INTRODUCTION

*Andrews (1979)* measured crown facial prominence, torque, and tip values of 120 Americans with ideal occlusion. These measurements were then used to design specialized brackets for each tooth in preadjusted appliances.

Corresponding author
Yanqin Lu, 213031@csu.edu.cn

According to previous research, craniomaxillofacial hard and soft tissue morphology differs by race and ethnicity (*Hideki et al., 2007*; *Yan et al., 2011*). Race has also been shown to be related to dental and occlusal characteristics. Several researchers (*Currim & Wadkar, 2004*; *Lombardo et al., 2015*; *Watanabe & Koga, 2001*) have reported differences in facial prominence, in-out, angulation, and inclination among Japanese, Indian, African, Caucasian, American, and Chinese populations.

In theory, prescriptions, including in-out (different base thickness), tip, and torque, should be designed based on racial characteristics. For example, bracket base thickness is determined by crown prominence to obtain the required first order correction. However, differences in prominence, which equate to differences in base thickness, between racial groups likely affects first order alignment. Tip and torque differences between races are also related to anchorage control and aesthetic effect. Several orthodontists have suggested that modifying prescriptions according to differences in race will lead to improved occlusal and aesthetic treatment outcomes (*Currim & Wadkar, 2004*; *Lombardo et al., 2015*). However, few follow-up studies on preadjusted appliances based on racial dental characteristics have been reported.

Currently, popular prescriptions, such as Roth, MBT, and Andrews, are based on data from patients in Western countries who have normal occlusion. In an early study from our group, a 3D coordinate measuring machine was used to measure dental casts of Chinese patients with normal occlusion (*Yang & Zeng, 1998*). Compared with American patients who were measured by Andrews, these Chinese patients showed obvious differences in crown facial prominence, torque, angulation, and especially in the in-out between the lateral incisors and the canines, as well as between the second premolars and the first molars, and in the torque, tip values of the anterior teeth (*Yang & Zeng, 1998*). Subsequently, a conventional ligating appliance was designed (*Zeng & Gao, 2008*) based on these data (*Yang & Zeng, 1998*). This appliance was later developed into a self-ligating appliance with the same prescription and design concept.

Popular prescription appliances are widely used in China and around the world with good treatment outcomes. However, it remains unknown whether modifications to prescriptions based on racial dental characteristics can improve treatment outcomes for occlusal details.

The purpose of this study is to compare the effectiveness of a Chinese prescription with that of a popular Western prescription (Roth) in a sample of Chinese patients. The null hypothesis is that there is no significant difference between the treatment outcomes for Chinese patients treated with either Chinese or Roth prescription appliances.

## METHOD

This retrospective study was approved by the Institutional Review Board at the Xiangya Stomatological Hospital Central South University (No.20190018). The production and clinical use of the appliance based on Chinese characteristics was approved by the China Food and Drug Administration. All participants gave written informed consent for inclusion in this study. All patients who were considered for this study received and

completed orthodontic treatment from the same physician at Xiangya Stomatological Hospital between 2015 and 2018.

## Inclusion and exclusion criteria

The inclusion criteria were as follows: (1) the patient had received comprehensive orthodontic treatment with either a Chinese (Z2, 3B Ortho) or a Roth (In-Ovation, Dentsply) prescription self-ligating appliance, both of which are 0.022 inch systems; (2) the patient underwent bonding between 12 and 30 years of age; (3) complete clinical records were available for the patient. The exclusion criteria were: (1) the patient showed craniofacial anomalies or syndromes or underwent orthognathic surgery; (2) the patient had congenitally missing teeth, a tooth deformity, or a fused tooth; (3) the patient was debonded early and showed poor compliance.

## Sample size calculation

The American Board of Orthodontics (ABO) Cast-Radiograph Evaluation (CR-Eval) was used to evaluate treatment outcomes. In accordance with several studies (*Detterline et al., 2010*; *Mislik et al., 2016*), a difference of five points in the mean ABO CR-Eval score was considered clinically significant in this study. The standard deviation of the ABO CR-Eval score was estimated to be 8.8 according to a previous study with a large sample size (*Cansunar & Uysal, 2014*). The alpha value and power were set at 0.05 and 80%, respectively. PASS software 11.0 was used to calculate the required sample size, with the results showing that 27 patients were needed for each group in this study. So we included 30 patients for each group.

## Propensity score matching

After inclusion and exclusion criteria were considered, two patients were excluded because their low compliance and early debonding may affect the results. The initial sample consisted of 91 patients, including 36 patients treated with a Chinese prescription, and 55 patients treated with a Roth prescription. Detailed data regarding the Chinese and Roth prescriptions are shown in Table 1.

Propensity score matching (PSM) was used to select the final sample from the initial sample. To prevent several covariates, including sex, age, malocclusion severity, and tooth extraction, from affecting the results, the ABO Discrepancy Index (DI) was used to quantify the severity of malocclusion for every patient. The DI considers overjet, overbite, anterior open bite, lateral open bite, crowding, occlusal relationship, lingual posterior crossbite, buccal posterior crossbite, cephalometric items, and other items (*Deguchi et al., 2005*). The initial sample was separated into three severity levels according to pre-treatment DI: low (DI < 7), medium (DI 8–16) and high complexity (DI ≥17) (*Cansunar & Uysal, 2014*).

In this study, the cervical vertebral maturation (CVM) method, which separates patients into 6 stages based on a study *Baccetti, Franchi & McNamara Jr (2005)*, was used to quantify the age of the patients according to pre-treatment cephalometric radiographs. Sex of the patients as well as non-extraction or extraction of premolars in the treatment plan were recorded as binary variables. DI, sex, extraction plan, and CVM stage were analysed and recorded for each patient in the two groups.

**Table 1** Data of the Chinese prescription and the Roth prescription self-ligating appliance.

| | Chinese prescription | | Roth prescription | |
| --- | --- | --- | --- | --- |
| | Torque | Angulation | Torque | Angulation |
| U1 | 11 | 4 | 12 | 5 |
| U2 | 7 | 6 | 8 | 9 |
| U3 | −3 | 7 | −2 | 13 |
| U4 | −7 | 2 | −7 | 0 |
| U5 | −7 | 4 | −7 | 0 |
| L1/L2 | 0 | 0 | −1 | 2 |
| L3 | −3 | 0 | −11 | 7 |
| L4 | −15 | 3 | −17 | −1 |
| L5 | −23 | 4 | −22 | −1 |

**Notes.**

The unit is degree (°).

A multivariable logistic regression model was calculated using SPSS software for Windows (version 25.0; IBM, Armonk, NY). For the propensity score analysis, prescription type was used as the dependent variable, while CVM stage (Cervical Stage, CS1, CS2, CS3, CS4, CS5, CS6), sex, DI severity level (low, medium, high), and extraction plan (Yes or No) were modelled as covariates. A one-to-one nearest neighbour matching algorithm was applied using SPSS software, and a calliper with a width of 0.02 times the standard deviation of the logit of the propensity score was applied as the matching criteria (*Xiao et al., 2018*).

After PSM, a final sample of 60 patients was obtained: 30 patients in the Chinese prescription (group 1) and 30 patients in the Roth prescription group (group 2). The baseline data of the two groups were compared using $\chi^2$-tests to ensure that the covariates were balanced.

## Treatment procedure

Direct bonding was used in this study. All brackets were bonded to the facial axis point of the tooth by the same physician. The wire sequence for all patients in this study was: 0.014-inch super elastic nickel-titanium, 0.016-inch super elastic nickel, 0.016-inch stainless steel, 0.018 × 0.025-inch super elastic nickel-titanium, and 0.019 × 0.025-inch stainless steel. Sliding mechanics was used to close the space for cases requiring premolar extraction. Intermaxillary elastic traction was avoided as far as possible. If a bracket got loose and lost, it would be replaced by a new one. All patients were asked to maintain good oral hygiene. All cases completed treatment with straight wires, and no extra wire bending was made throughout the whole clinical process. Impressions were taken to obtain finishing dental casts on the day of debonding.

## Treatment evaluation

In this study, the ABO CR-Eval was selected as the evaluation index to quantify treatment outcomes. The ABO CR-Eval consists of eight measurement items: alignment, marginal ridges, buccolingual inclination, occlusal relationships, occlusal contacts, overjet,

interproximal contacts, and root angulation. Post-treatment plaster casts and panoramic X-rays were analysed using an ABO calibration kit. In order to compare the torque control of the two prescriptions, lateral cephalograms were used to measure U1/SN for upper incisors and L1/MP for lower incisors.

To minimize the error between investigators, the same types of measurements were performed by one investigator. Before ABO measurements were taken, the investigator was trained using ABO instructions and tutorial videos from the ABO website. All the casts and radiographs were assigned a random coded number by an assistant to blind the investigator to the experimental conditions.

### Reliability

The validity of the ABO CR-Eval for assessing treatment outcomes of Chinese patients has been established in a previous study (*Song et al., 2013*). To train for ABO CR-Eval, there are 3 sets of calibration dental casts with collective, agreed-upon scores. To reduce systematic bias, investigators should guarantee that measured ABO scores are consistent with the established measurements of these three casts. To assess the systematic errors, 20 patients were selected at random and another examiner who had also received training for ABO measured the scores again in comparison with the investigator in this study. Bland-Altman plots were used to assess systematic errors by the Graphpad Prism 7 (GraphPad Software Inc, San Diego, CA, USA). To assess intra-examiner reliability and reproducibility in the current study, measurements by the investigator for 15 randomly selected cases were taken one month after the first measurements, and the intraclass correlation coefficient (ICC) was calculated.

### Statistical analysis

Independent samples t-tests were performed to compare the total score of the ABO CR-Eval as well as the sub-scores between group 1 and group 2. Because of differences in the in-out design between the two prescriptions, alignment scores were compared for each tooth type. For each individual patient, the scores for the same tooth type on the left and right sides of the jaw were combined. Independent samples t-tests were also performed for U1/SN and L1/MP. P-values of less than 0.05 were considered to be statistically significant.

## RESULTS

### Error of the method

The ICC values for the CVM stage evaluations, the DI scores, and the ABO scores were 0.92, 0.97, and 0.98, respectively. The ICC values in this study were above 0.8, suggesting good consistency and reliability. The Bland-Altman plot (Fig. 1) show that the mean difference between the score of examiner (S1) and the score of another examiner (S0) was −0.15, with limits of agreement (mean ± 1.96 SD) of −2.55 to 2.25.

### Final sample

PSM and baseline data of the final sample are provided in Table 2. Most cases were classified as either medium or high severity according to their DI score. The χ2 tests showed that
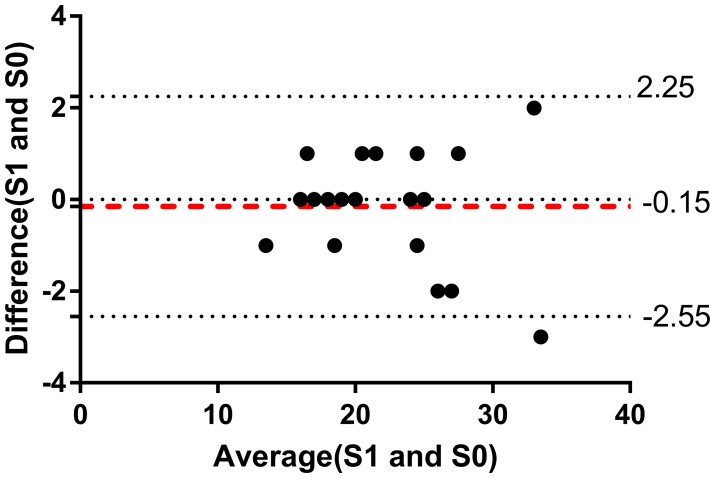

**Figure 1** **Bland-Altman plot of ABO score in this study.** Bland-Altman plot of ABO score by the investigator (S1) and ABO score by another examiner (S0).

the distributions of CVM stage, DI score, sex, and extraction plan were not significantly different between the two groups (p >0.05). In the final sample, the anterior Bolton index were 79.4% ± 1.4% for Chinese group and 79.1% ± 1.3% for Roth group. The total Bolton index were 91.6% ± 0.9% for Chinese group and 91.5% ±0.9% for Roth group. The treatment duration was 22.4 ± 3.6 months for the Chinese group and 23.5 ± 3.0 months for the Roth group.

### ABO CR-Eval scores and incisors inclination

The ABO CR-Eval total scores and the eight sub-scores are provided in Table 3. The results show that there were no significant differences in total ABO score or in the sub-scores, including marginal ridges, buccolingual inclination, occlusal relationships, occlusal contacts, overjet, interproximal contacts, and root angulation between the two groups. However, a statistically significant difference in alignment score was observed between group 1 and group 2.

The alignment score for each tooth is shown in Table 4. The only significant difference found between the groups was the alignment score for the mandibular canine. There was a lower mean score for the mandibular canine in group 1 (the Chinese prescription group) compared to that of group 2 (the Roth prescription group). According to the ABO CR-Eval analysis of group 2, most of the alignment score for the mandibular canine was due to a mandibular canine lingual deviation from the ideal arch line. As shown in Fig. 2, the mandibular canines were located at the ideal position in the dental arch for patients with the Chinese prescription (Fig. 2D), while the mandibular canines were not arranged in a smooth curving line and deviated lingually from the ideal position for the Roth prescription (Fig. 2B).

The incisor inclination including U1/SN and L1/MP of the two groups is shown in Table 5. The statistical analysis showed no significant differences in U1/SN and L1/MP between the two groups.

**Table 2  Baseline data of the two groups of patients after PSM.** The table shows the number of patients in each category.

|  | Baseline data after PSM | | P value |
|---|---|---|---|
|  | Chinese prescription | Roth prescription |  |
| Sex | | | |
|     male | 14 | 15 | 0.796 |
|     female | 16 | 15 | 0.796 |
| Age | | | |
|     CVM I-III | 1 | 1 | 1 |
|     CVM IV | 9 | 7 | 0.559 |
|     CVM V | 14 | 16 | 0.606 |
|     CVM VI | 6 | 6 | 1 |
| Complexity | | | |
|     DI low | 2 | 2 | 1 |
|     DI medium | 11 | 12 | 0.791 |
|     DI high | 17 | 16 | 0.795 |
| Extraction plan | | | |
|     Extraction | 5 | 6 | 0.739 |
|     Non-extraction | 25 | 24 | 0.739 |

Notes.
$\chi^2$-test.
*$p < 0.05$, statistical significance was set at $p < 0.05$.

**Table 3  Independent samples *t*-test for the eight ABO CR-EVAL items and the total ABO score.**

| Type | Mean | | Stand deviation | | Mean difference | 95% Confidence interval | | P value |
|---|---|---|---|---|---|---|---|---|
|  | Chinese | Roth | Chinese | Roth |  | Lower | Upper |  |
| Total ABO score | 22.03 | 23.87 | 4.82 | 5.26 | −1.83 | −4.44 | 0.78 | 0.165 |
| Alignment | 4.70 | 6.13 | 1.60 | 2.10 | −1.43 | −2.40 | −0.47 | 0.004* |
| Marginal Ridges | 3.70 | 3.93 | 1.86 | 1.39 | −0.23 | −1.08 | 0.61 | 0.584 |
| Buccolingual inclination | 3.40 | 3.27 | 1.69 | 1.96 | 0.13 | −0.81 | 1.08 | 0.779 |
| Overjet | 3.87 | 4.10 | 2.37 | 1.95 | −0.23 | −1.36 | 0.89 | 0.679 |
| Occlusal contacts | 2.77 | 3.33 | 1.96 | 2.02 | −0.57 | −1.60 | 0.46 | 0.275 |
| Occlusal relationships | 2.87 | 2.57 | 2.73 | 2.56 | 0.30 | −1.07 | 1.67 | 0.662 |
| Interproximal Contacts | 0.00 | 0.03 | 0.00 | 0.18 | −0.03 | −0.10 | 0.30 | 0.321 |
| Root angulation | 0.73 | 0.50 | 0.69 | 0.62 | 0.23 | −0.11 | 0.58 | 0.177 |

Notes.
Independent samples t-test.
*$p < 0.05$, statistical significance was set at $p < 0.05$.

# DISCUSSION

Propensity score analysis has been widely used in many recent clinical studies to compare the therapeutic effects and clinical prognoses of different treatment plans (*Wu et al., 2019*; *Zeng et al., 2019*). In orthodontic clinical studies, many confounding factors can affect clinical results, and these have often been overlooked. In this study, PSM was used to adjust

**Table 4  Independent samples t-test for the alignment score of tooth types by two appliances.**

| Type | Mean | | Stand Deviation | | Mean Difference | 95% Confidence Interval | | P value |
|------|---------|------|---------|------|------|------|------|------|
| | Chinese | Roth | Chinese | Roth | | Lower | Upper | |
| U1 | 0.23 | 0.10 | 0.50 | 0.31 | 0.13 | −0.08 | 0.35 | 0.220 |
| U2 | 0.23 | 0.30 | 0.50 | 0.53 | −0.07 | −0.34 | 0.20 | 0.621 |
| U3 | 0.33 | 0.60 | 0.55 | 0.77 | −0.27 | −0.61 | 0.08 | 0.127 |
| U4 | 0.53 | 0.67 | 0.73 | 0.88 | −0.13 | −0.55 | 0.29 | 0.527 |
| U5 | 0.47 | 0.20 | 0.73 | 0.43 | 0.23 | −0.07 | 0.54 | 0.137 |
| U6 | 0.40 | 0.33 | 0.50 | 0.55 | 0.07 | −0.20 | 0.34 | 0.623 |
| U7 | 0.40 | 0.60 | 0.56 | 0.77 | −0.20 | −0.55 | 0.15 | 0.256 |
| L1 | 0.03 | 0.10 | 0.18 | 0.31 | −0.07 | −0.20 | 0.06 | 0.309 |
| L2 | 0.20 | 0.27 | 0.41 | 0.58 | −0.07 | −0.33 | 0.19 | 0.610 |
| L3 | 0.20 | 1.03 | 0.41 | 1.03 | −0.83 | −1.24 | −0.43 | 0.000* |
| L4 | 0.20 | 0.13 | 0.48 | 0.35 | 0.07 | −0.15 | 0.28 | 0.542 |
| L5 | 0.17 | 0.20 | 0.46 | 0.48 | −0.03 | −0.28 | 0.21 | 0.786 |
| L6 | 0.80 | 0.63 | 0.96 | 0.99 | 0.17 | −0.34 | 0.67 | 0.513 |
| L7 | 0.63 | 0.80 | 0.81 | 0.89 | −0.17 | −0.61 | 0.27 | 0.450 |

**Notes.**

U1–U7: maxillary central incisor to second molar, L1–L7: mandibular central incisor to second molar.

*$p < 0.05$, statistical significance was set at $p < 0.05$.

the baseline data and balance confounding factors, allowing for a more direct comparison of treatment outcomes according to prescription type.

Many occlusal indices are used to evaluate therapeutic effects, such as the peer assessment rating (PAR); the index of complexity, outcome and need (ICON); and the ABO Cast-Radiograph Evaluation (CR-Eval). According to *Cansunar & Uysal (2014)*, the PAR and the ICON are not sufficient to accurately and precisely measure tooth position. The ABO CR-Eval is generally used in clinics to improve treatment outcomes and as a measurement tool for scientific research.

In the current study, the total ABO score was $22.03 \pm 4.82$ for the Chinese prescription appliance and $23.87 \pm 5.26$ for the Roth prescription appliance. In a previous study by *Brown et al. (2015)*, ABO CR-Eval scores were $28.5 \pm 8.5$, $32.3 \pm 7.8$, and $32.3 \pm 9.3$ for three different self-ligating appliances. Detterline's (*Detterline et al., 2010*) experimental results showed that scores following treatment with a 0.018-inch bracket and a 0.022-inch bracket were $26.3 \pm .0.0$ and $28.5 \pm 21.3$, respectively. Thus, compared with the scores reported in other studies, the Chinese patients in the current study treated with either the Chinese or the Roth prescription achieved satisfactory total scores. As the Roth prescription appliance is one of the most effective and popular appliances in clinical use, the similar scores between the Chinese prescription group and the Roth prescription group suggest that both prescriptions are effective for overall treatment outcomes as well as the outcomes of the above-mentioned seven items.

In our study, the top four items with the highest scores, listed in descending order, were alignment, marginal ridges, buccolingual inclination, and overjet for the Chinese prescription group and alignment, overjet, marginal ridges, and occlusal contacts for the

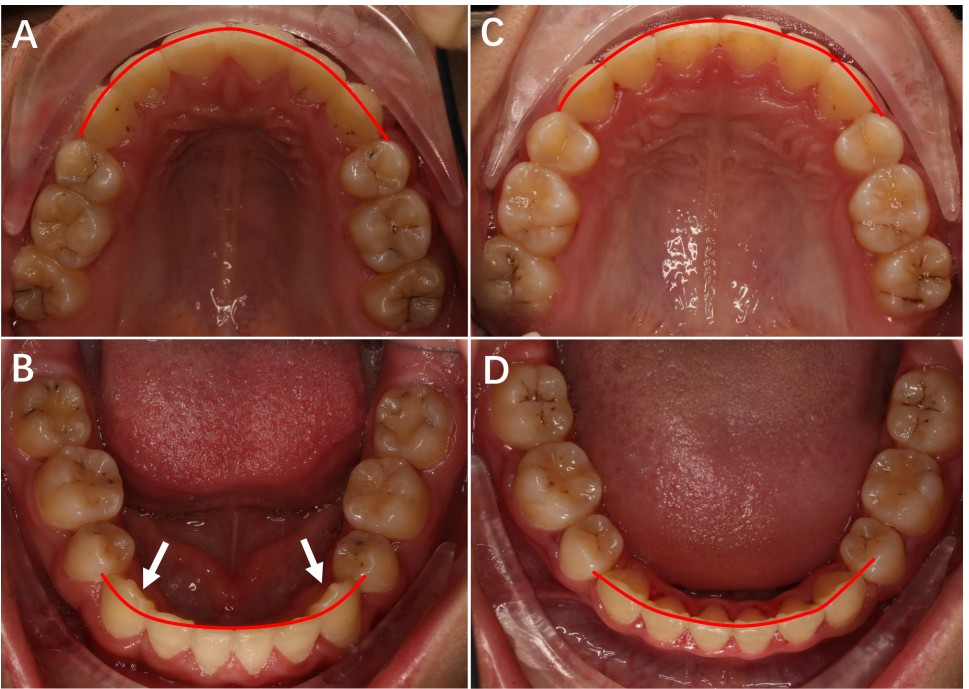

**Figure 2 The upper and lower occlusal pictures of patients treated with Roth prescription and Chinese prescription appliance.** (A) The upper occlusal picture of Roth prescription. (B) The lower occlusal picture of Roth prescription. (C) The upper occlusal picture of Chinese prescription. (D) The lower occlusal picture of Chinese prescription.

**Table 5 Independent samples t-test for the upper and lower incisors inclination of two groups of patients.**

| Type | Mean | | Stand Deviation | | Mean Difference | 95% Confidence Interval | | P value |
|------|---------|------|---------|------|------|-------|-------|---------|
| | Chinese | Roth | Chinese | Roth | | Lower | Upper | |
| U1/SN | 105.1 | 106.0 | 7.4 | 6.9 | −0.8 | −4.5 | 2.9 | 0.68 |
| L1/MP | 96.0 | 95.0 | 6.4 | 6.0 | 1.0 | −2.2 | 4.3 | 0.52 |

Notes.
Independent samples $t$-test.
$^\star p < 0.05$, statistical significance was set at $p < 0.05$.

Roth prescription group. In previous studies, the items with the highest scores were reported to be alignment (*Cook, Harris & Vaden, 2005*; *Yassir et al., 2018*), occlusal contacts (*Aszkler et al., 2014*), and occlusal relationships (*Okunami et al., 2007*). The item with the lowest score in this study was interproximal contacts, and this result was consistent with that of *Yassir et al. (2018)*, and *Aszkler et al. (2014)*. This finding suggests that closing the space is not very challenging for most orthodontic cases. In contrast, orthodontists should pay close attention to other items, such as alignment, marginal ridges, and occlusal relationships, from the beginning to the end of treatment.

The alignment scores of the mandibular canines in the Chinese prescription group were significantly lower than those in the Roth prescription group. In a study by *Yang & Zeng (1998)*, the discrepancies in prominence between the maxillary lateral incisor and the
maxillary canine, the mandibular lateral incisor and the mandibular canine, the maxillary second premolar and the maxillary first molar, and the mandibular second premolar and the mandibular first molar of Chinese individuals with normal occlusion were considerably different from those of American patients who were measured by Andrews. Due to these differences in crown prominence, there exist differences in the bracket thicknesses between the two prescriptions. Because the prominence discrepancies between the mandibular canine and the lateral incisor of the Chinese patient group were greater than those of the American patients measured by Andrews, the Roth prescription base thickness may lead to the mandibular canines deviating from the ideal buccal-lingual position in Chinese patients treated with the Roth appliance. In addition, the ideal torque of the mandibular canine for Chinese patients is $-3°$ (*Yang & Zeng, 1998*), but the Roth's torque prescription value is $-11°$, meaning that the Roth prescription appliance produces excessive lingual-crown torque for Chinese patients. These differences can explain why the Chinese prescription group achieved better alignment of the mandibular canine than the Roth group did in our study. In contrast, the in-out and torque value differences were not as large for the other teeth as they were for the mandibular canines. This is the most likely explanation for the absence of significant differences in the alignments of the other teeth. Only misalignments of greater than 0.5 mm were scored by the ABO CR-Eval, and it is difficult to accurately measure discrepancies below 0.5 mm using the ABO gauge and the naked eye. Theoretically, real full-size arch wires can give more expression of the bracket prescriptions, however, it is not realistic clinically because there still exists clearance between finishing arch wire and the bracket slot. In this study, both groups were finished with the same arch wire, and thus the interaction between finishing arch wire and bracket slot was comparable.

In this study, satisfactory treatment outcomes were achieved by the Chinese prescription appliance for Chinese patients, and the Chinese prescription had the advantage of avoiding lingual malposition of the mandibular canine and thus showed improved alignment compared to the Western prescription. It might be possible that customized brackets will solve the problem of dental anatomical differences between patients in the future. However, at the moment customized brackets are more expensive and the production of an individualized bracketr for each patient takes time. Preadjusted appliances based on mean values of normal occlusion are absolutely the most popular brackets now. The purpose of this study was not to create a "fit for all" appliance. Sometimes bending wires cannot be avoided, but prescriptions based on Chinese normal occlusion may reduce wire bending for the mandibular canines of a significant number of Chinese patients.

Our results have significance for other ethnic groups as well. The dental characteristics of a sample of patients with normal occlusion should be measured for each ethnic group. If there exist significant differences between individuals of different races, it is better to modify the bracket prescriptions based on racial dental characteristics. The resulting overall treatment outcomes may not demonstrate obvious improvements, but the position of certain teeth can be controlled more precisely, closer to normal occlusion. Furthermore, if patients must be treated with appliances which were developed using data from individuals of other races, orthodontists should pay special attention to the positions of certain teeth with regards to dentition and torque magnitudes in the finishing stage. Sometimes

additional wire bending of the first, second, and third order are necessary to realize proper alignment, occlusion, and aesthetic effects for certain teeth.

## Limitations

There are several limitations to this study. This was a retrospective study which means there are limitations with respect to patient selection and non-randomization. A prospective study will be performed in the future to both explore and revise the Chinese prescription appliance.

Direct bonding of brackets was performed in this study. The indirect bonding technique may be more rigorous. Yet the direct bonding technique is the most common and popular technique in clinics around the world. Moreover, a recent paper (*Li et al., 2019*) showed that there were no significant differences in the mean errors by the two bonding techniques.

All the participants were treated by one clinician in this study, which increases the internal validity and may affect the external validity. However, different clinicians have different treatment preferences, even different levels of clinical skills. If patients from different clinicians had been included in this retrospective study, it would be hard to compare the treatment outcomes of two prescriptions and clinician could have been a confounder.

The angulation and torque values in the completed cases could not be directly measured by the ABO CR-Eval. To compare the degrees of angulation and magnitudes of torque, it may be necessary to use customized measurement software to measure these values with 3D digital models in a follow-up study.

## CONCLUSIONS

There were no significant differences in the total ABO CR-Eval score or the scores of seven other subitems between Chinese patients treated with a Chinese prescription and those treated with a Western prescription appliance. However, there was a significant difference in the alignment score of the mandibular canine between the two groups, with the Chinese self-ligating prescription group showing a better score than the Western prescription appliance group.

## ACKNOWLEDGEMENTS

We thank Professor Xianglong Zeng for his help in this study.

### Funding

The authors received no funding for this work.

### Competing Interests

The authors declare there are no competing interests.

## Author Contributions

- Yanhao Chu conceived and designed the experiments, performed the experiments, authored or reviewed drafts of the paper, and approved the final draft.
- Lingling Zhang performed the experiments, prepared figures and/or tables, authored or reviewed drafts of the paper, and approved the final draft.
- Yatao Zhao analyzed the data, authored or reviewed drafts of the paper, and approved the final draft.
- Fang Yi performed the experiments, prepared figures and/or tables, and approved the final draft.
- Yanqin Lu conceived and designed the experiments, authored or reviewed drafts of the paper, and approved the final draft.

## Ethics

The following information was supplied relating to ethical approvals (i.e., approving body and any reference numbers):

The Xiangya Stomatological Hospital,Central South University granted Ethical approval to carry out the study (No. 20190018).

## Data Availability

The raw measurements are available as a Supplemental File.

## Supplemental Information

Supplemental information for this article can be found online at http://dx.doi.org/10.7717/peerj.10605#supplemental-information.

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
