# Peer review of "Effectiveness of modifications to preadjusted appliance prescriptions based on racial dental characteristics assessed by the ABO Cast-Radiograph Evaluation: A propensity score matching study"

_PeerJ, doi:10.7717/peerj.10605_

## Round 0.1 · original submission · Major Revisions

An important issue that has been raised by the reviewers is (1) the lack of an a priori sample size calculation. Furthermore (2) the reliability analysis should be extended. As (3) expression of torque by a certain bracket prescription is the key issue in the variation of shape of the incisors between races I would like to see more clearly how this issue is addressed in the scoring systems you use. I also miss s discussion of (4) the limitations of the study. Please consider all comments of the reviewers carefully as they help to improve your manuscript.

Finally I want to support the remarks of several of the reviewers about the need of improving the writing. I have seen that an editing service has looked at the manuscript but there are still many grammatical errors. The manuscript also needs editing from a professional academic editing service familiar with the orthodontic field to improve the readability.

Reviewer 1 ·

Basic reporting

The paper was well written and well organized. Sufficient literature was provided.

Experimental design

The Authors used the CVM method by Baccetti et al 2002. That method consisted of 5 stages and used Roman numerals. The authors reported 6 stages. Please clarify.
Was bracket placement standardized. Was a direct or indirect bonding technique used? Please report the wire sequence.

Validity of the findings

The methodology of the study was correct. please report the 95% confidence interval in Tables III and IV.

Reviewer 2 ·

Basic reporting

English language need a complete revision, because it is not conform to the professional standards of expression and the overall style is not good.
The language is not very fluent and often appears as a literal translation.
Corrections must be made in all the parts of the paper.

Litterature references is sufficient and appropiately referenced.

Introduction section introduces the problems covered by the study in a fairly clear and detailed way

As far as the background is concerned, it should be better explained the development of chinese prescription preadjusted appliance

Article is structured correctly and follows author guidelines

Figures have sufficient resolution but the pictures of mandibular occlusal view, both in roth and chinese prescription patients, have been acquired incorrectly and do not allow a good view of the arch

Tables are well organized, but the footnotes must better describe the data reported

Experimental design

The paper is a research article which evaluates a rather specific aspect in the field of orthodontics. It can fall within the aims and scope of PeerJ, even if, considering its high specificity, it could be of interest to a minority part of the readers

The purpouse of the study is interesting,and the paper provides an interesting figure that contributes, even if not to a significant extent,to filling a knowledge gap in the field of orthodontics

Investigation procedures shown some important limits:
1- Indirect brackets bonding procedure was to be preferred as a more rigorous positioning of the bracket was required
2- A standard bolton index is not considered in the inclusion criteria, in order to exclude any dental mass anomalies in patients sample

Validity of the findings

In the results section must be indicated the single values of intraclass correlation coefficient (ICC) and not generally report that were all greater than 0,90.
The findings of the study are sufficiently interesting , well discussed and adequately supported by litterature references

Additional comments

The paper is a research article which evaluates a rather specific aspect in the field of orthodontics. providing an interesting figure that contributes, even if not to a significant extent,to filling a knowledge gap.
Some methodological bias must be corrected and an extensive english language revision is necessary before the paper can be considered for pubblication

·

Basic reporting

The article has covered all sections in the background with satisfactory literature references. However, the grammatical errors makes it difficult to understand.

Hypothesis is not stated.

Data from tables are readable and sufficient.

Result section does not address the aims of the study entirely.

Experimental design

Research question vague.

Methodology sound but bias still exists inherent to the study design.

Lack of sample size calculation a serious concern.

Validity of the findings

Data provided robust, statostically sound. Propensity score match increases the valididty of the findings. However, lack of sample size calculation and superficial reliability assessment undermine the overall finidngs.

Additional comments

Thank you for asking me to review the manuscript titled ‘Effectiveness of modifications to pre-adjusted appliance prescriptions based on racial dental characteristics assessed by ABO cast-radiograph evaluation - A propensity score matching study’. The study compares the Chinese bracket prescription with Roth prescription. The authors have attempted to eliminate the confounders by carrying out a propensity score match between the groups. Although the authors have carried out a PSM for reducing the effect of confounders, inherent problems with study design still exists.

Major concerns:

The main concern I have in this study is the lack of sample size calculation. Additionally, there is discrepancy in the reporting of overall sample. The method section has no clear information on the number of participants included. The abstract reports 91 participants and Table 2 reports 60 participants.

The study is reported as a Cohort design but the write up feels retrospective. If the study is prospectively carried out, did the authors register the protocol in any public database?

One clinician treated all the included participants. Although this increases the internal validity, the external validity is lost.

The authors failed to evaluate the torque at the end of treatment. The objective of the study is ‘effect of prescription’ and it feels incomplete without torque assessment.

The authors finished treatment on 19x25 SS. The full expression of tip and torque does not happen on this archwire, as the play is high. Perhaps the lack of difference is because of the failure to use a full size archwire?

Following variables during treatment would make an impact on the final score; further information is required on the use of elastics, number and type of breakages, oral hygiene, space closure mechanics used, etc.

When were impressions taken post debond, on the same day? Were bonded retainers fitted on the same day as debond?

Information on bracket slot dimension including company name, archwire sequence and overall duration of treatment should be provided.

Reliability test was carried out using ICC. Bland and Altman test is required in addition to ICC to test both random and systemic errors.

The authors need to check the manuscript carefully for grammatical errors.

Reviewer 4 ·

Basic reporting

The manuscript is clearly written and the hypotheses is coherent with the introduction.

Experimental design

The research deals with a topic of limited interest. The research question is well defined but not very relevant. It is difficult to identify which knowledge gap is filled.

The methods are well described, even if the retrospective design of the study should be better underlined.

Validity of the findings

Validity of findings is limited for the possible selection bias due to the retrospective design and the exclusion of all the patients with poor compliance.

Conclusions are well stated and linked to original research question, but they overestimate the finding.

Additional comments

The article deals with a topic that is not very interesting since customization of brackets according to single patient need is possible. The importance of brackets prescription on the outcome of an orthodontic treatment is limited since information might not be fully expressed and in complex cases teeth position is more influenced by side effects of treatment mechanics than bracket prescription. Furthermore, the retrospective design reduces the validity of the study.

---

## Round 0.2 · Major Revisions

The manuscript has been improved considerably after the comments of the referees. However, there is still room for improvement. Overall you have answered the comments of the referees satisfactorily, but unfortunately, many of the answers are not included in the revised text. The readers may have the same questions/comments, so it is important to elucidate those points in the revised manuscript.

In the attached file with annotations, I have indicated where you could include the points that are important to add. I have used your revised ms for this and accepted nearly all changes.

Furthermore, Bland Altman plots should be added as indicated by 2 of the 4 referees, and this is also my recommendation.

Finally, about figure 1. This figure is not mentioned in the text, but I assume you want to use the new figure of the models? please add.

Reviewer 1 ·

Basic reporting

The Authors replied appropriately to the issues raised.

Experimental design

The Authors replied appropriately to the issues raised.

Validity of the findings

The Authors replied appropriately to the issues raised.

Additional comments

The Authors replied appropriately to the issues raised.

---

## Round 0.3 · Minor Revisions

Your manuscript is entering the final stage. I have made editorial changes in the attached word-file and I have two remarks that need to be solved (see manuscript). Please accept all changes if you agree and make changes according to the two remarks.

---

## Round 0.4 · accepted · Accept

I have accepted all changes and corrected some spelling errors. A pdf of the final manuscript is attached.